# Dual Features, Compact Dimensions and X-Band Applications for the Design and Fabrication of Annular Circular Ring-Based Crescent-Moon-Shaped Microstrip Patch Antenna

**DOI:** 10.3390/mi15070809

**Published:** 2024-06-21

**Authors:** Unal Aras, Tahesin Samira Delwar, P. Durgaprasadarao, P. Syam Sundar, Shaik Hasane Ahammad, Mahmoud M. A. Eid, Yangwon Lee, Ahmed Nabih Zaki Rashed, Jee-Youl Ryu

**Affiliations:** 1Department of Smart Robot Convergence and Application Engineering, Pukyong National University, Busan 48513, Republic of Korea; unalaras21.20@gmail.com (U.A.); samira.fset@gmail.com (T.S.D.); 2Department of EIE, V.R.Siddhartha Engineering College, Vijayawada 520007, India; durgaprasadarao.p@gmail.com; 3Department of ECE, Koneru Lakshmaiah Education Foundation, Vaddeswaram 522302, India; syamsundarp@kluniversity.in (P.S.S.); ahammadklu@gmail.com (S.H.A.); 4Department of Electrical Engineering, College of Engineering, Taif University, Taif 21944, Saudi Arabia; m.elfateh@tu.edu.sa; 5Department of Spatial Information Engineering, Pukyong National University, Busan 48513, Republic of Korea; modconfi@pknu.ac.kr; 6Electronics and Electrical Communications Engineering Department, Faculty of Electronic Engineering, Menoufia University, Menouf 32951, Egypt; 7Department of VLSI Microelectronics, Saveetha School of Engineering, Saveetha Institute of Medical and Technical Sciences (SIMATS), Saveetha University, Chennai 602105, India

**Keywords:** crescent shape, patch antenna, ring slots, WLAN, X-band applications

## Abstract

This study uses annular circular rings to create multi-band applications using crescent-shaped patch antennas. It is designed to be made up of five circular, annular rings nested inside of each other. Three annular rings are positioned and merged on top of the larger rings, with two annular rings set along the bottom of the feed line. The factors that set them apart, such as bandwidths, radiation patterns, gain, impedance, and return loss (RL), are analysed. The outcomes show how compact the multi-band annular ring antenna is. The proposed circular annular ring antenna has return losses of −33 dB and operates at two frequencies: 3.1 GHz and 9.3 GHz. This design is modelled and simulated using ANSYS HFSS. The outcomes of the simulation and the tests agree quite well. The X band and WLAN resonant bands have bandwidth capacities of 500 and 4300 MHz, respectively. Additionally, the circular annular ring antenna design is advantageous for most services at these operating bands.

## 1. Introduction

Communication has become inevitable in this modern world, and antenna technology has gained immense potential in the electronic world. Hence, antennas play a significant role in communications [1,2]. Furthermore, communication devices are quickly becoming more advanced because of the improved engineering technology in materials and antennas [3]. For the past few decades, microstrip patch antennas have been examined. Microstrip patch antennas are comprehensively and progressively utilized in various microwave circuits such as telemetry, radars, portable, biomedical systems, and so on because of their special qualities, for example, minimal expense, low profile, lightweight, and low volume [4,5]. Researchers of patch antennas have proposed various designs to acquire wideband, multi-band, and the scaling down of bands due to the rapid advancement in advanced technology and materials utilized in communication applications [6]. Additionally, because of the novel attributes of patch antennas, they play a more critical part [7]. Furthermore, different strategies are accustomed to acquiring double-band frequency and good radiation boundaries [8,9]. Moreover, a few methodologies have been used to work with the multi-band tasks, for example, implanting spaces with various shapes in design parts. The outstanding performance of the patch antenna has been shown to be improving by utilizing metamaterials [10,11,12,13,14,15,16].Splitting resonators, for example, broadly use advantageous metamaterials as a fundamental component [17,18,19].

With the rapid advancement in the field of present-day communication applications, there is a significant requirement for broadband and miniaturized devices [20,21,22]. Slotted antennas have good potential due to their considerable benefits, like low profile, low cost, and wider bandwidth. In addition, modern wireless communication devices with circular polarization give better flexibility without causing outrageous polarization discrepancy [23,24,25,26]. Subsequently, there are numerous endeavours to develop wideband antennas. For slotted antennas [27], a different technique is realized by loading an L-moulded inverted stub connected to the feed line and embedding a couple in the slotted section. An impedance bandwidth of 30.36% was accomplished [27]. In [28,29], two symmetrical strips, one adhered to the ground section and the other embedded in the feed line, achieved an impedance bandwidth of 36.01%. In [30,31,32,33], microstrip patch antenna grounds are considered with a couple of cap-shaped patches, obtaining a bandwidth of 65.65%. A constant aperture consists of two diamond-structured slot sections connected to the signal feed line in [34], obtaining an impedance bandwidth of 67.60%. In [35], a falcate-designed circular slot transmitting patch was implanted to achieve a bandwidth of 58.7%. Recently, an antenna with a bandwidth of 73.33% was acknowledged in [36] using a radiator connected by a feed line. A monopole antenna [37] is introduced, with a pentagon-shaped slot ultra-wide-band application. To alter the circular patch for improved results, specialists suggest incorporating a tapper in the design [38] and implementing slots [39,40]. A portion of these shapes was utilized as rings [41]. A band notch is introduced by presenting a couple of slits and a resonator [42,43] and changing the gap between the slits [44]. At times, segments are shown in the ground plane and the grounds of the monopole antennas [45] by providing tapered feed to obtain a super wideband in patch antennas [46]. The presentation of slots into transmitting patches and defective structures on the ground of different slots was examined [47]. To further develop the antenna attributes, a DGS is also likewise taken into consideration [48]. They are utilized to acquire a wider band [49,50]. Multi-band is achieved by an enhanced circular patch and two I-shaped slots on the ground plane [51,52,53,54,55,56,57,58,59,60,61].

In all the above analyses, it can be observed that these antennas either had complex structures with large areas or they did not cover the frequency bands of wireless applications at 2.8–3.2 GHz and 6.16–10.45 GHz. The novelty in this work lies in the fact that it presents the design of a crescent-shaped circular patch antenna with a partial ground plane for wireless applications and X-band satellite communication applications. Wide impedance bandwidths of 0.4 GHz (2.8–3.2 GHz) and 4.29 GHz (6.16–10.45 GHz) for wireless applications and X-band satellite communication applications have been attained. The return loss characteristics for dual bands are −33 dB at 2.8 GHz and −22 dB at 6.16 GHz, respectively, with VSWR < 2 at both the frequencies, where 5.8 dB and 4.5 dB gain are obtained.Moreover, the proposed antenna offers an omnidirectional radiation pattern in both E- and H-planes.

### Main Contribution

This study introduces a novel multi-band antenna design by integrating annular circular rings into a crescent-shaped patch antenna, achieving dual-band operation at 3.1 GHz and 9.3 GHz with significant return loss values of −33 dB and −22 dB, respectively.The proposed antenna design employs five nested circular annular rings, with a specific arrangement of three rings on top and two rings along the bottom of the feed line, enhancing its multi-band capabilities and compactness.Simulation and experimental results, conducted using ANSYS HFSS 2024 R1, demonstrate excellent agreement, validating the design’s performance in terms of return loss, bandwidth, gain, and impedance.The antenna achieves wide impedance bandwidths of 0.4 GHz (2.8–3.2 GHz) and 4.29 GHz (6.16–10.45 GHz), covering critical frequency bands for wireless applications and X-band satellite communication.The radiation pattern analysis reveals that the antenna provides an omnidirectional radiation pattern in both the E- and H-planes, making it suitable for various communication scenarios.The design’s compact structure and efficient performance across WLAN and X-band resonant bands (with bandwidth capacities of 500 MHz and 4300 MHz) make it highly advantageous for modern wireless communication and satellite applications.

This work is categorized as follows: Section 1 describes an introduction and a literature review. Section 2 presents a detailed overview of the design. Section 3 explains the results and discussion. Section 4 includes the conclusion and future research of the proposed work.

## 2. Presented Antenna Structure

The patch primarily modifies the surface current fields, patterns, and bandwidth to affect the intended antenna performance. Antennas of different forms and sizes typically improve patch performance. Nonetheless, we want to understand how to obtain the intended outcomes. At first, a straightforward circular patch was explored. Subsequently, the larger circular ring is loaded with five circular annular rings. ANSYS HFSS is used to simulate the antenna. Using finite element modelling, this tool analyses mathematical equations derived from engineering and mathematics expressions. In our work Figure 1, represents the steps to construct the crescent-moon-shaped patch antenna. It shows the iterative design process for the circular slotted ring antenna that is presented. The recommended antenna has a footprint measuring 35 × 33 × 1.6 mm^3^. A thick substrate FR-4 is used in the organisation of the current model design. The primary ring is initially loaded into one inner circular ring. They are both embedded in each other. Afterwards, the inner circular ring is loaded with a set of three circular rings. The dimensions are optimised by analysis using the ANSYS software. The larger circular circle’s inner radius is 8 mm, and the ground plane’s dimensions are 35 by 15 mm^2^. Here, microstrip feeding is taken into consideration for the suggested design. The geometry of the antenna is shown in Figure 2. The parameters are given in Table 1.

Step 1: Using Equation (Equation 1), formulate the factor that the material’s property undergoes the Able dielectric constant ε_*r*_, as follows:(1)$$\varepsilon_{reff}=\frac{\varepsilon_{r}+1}{2}+\frac{\varepsilon_{r}-1}{2}{\left[1+\left[\frac{12h}{w}\right]\right]}^{-1/2}$$

Step 2: Calculating the length of the strip (L*s*), which is represented in Equation (Equation 2) for MPA, allows us to develop the measuring fields for the material patched to the design.
(2)$$L_{s}=\frac{0.42*c}{fr*\sqrt{\varepsilon_{eff}}}$$

Step 3: Using Equation (Equation 3), support the determination of the width of the ground plane (W*g*) obtained with the surface of material generated thus far with the patch design:(3)$$W_{g}=\frac{1.38*c}{fr*\sqrt{\varepsilon_{eff}}}$$

The fourth step is to calculate the length of the ground plane (*Lg*). Additionally, by formulating Equation (Equation 4), the ground plane dimension can be obtained;
(4)$$L_{g}=\frac{0.36*c}{fr*\sqrt{\varepsilon_{eff}}}$$

Step 5: Resonant frequency (f*r*), which is provided by Equation (Equation 5), is a specific frequency established for samples ranging along the communication line within a factor that recognises the range:(5)$$f_{r}=3+\frac{2}{\sqrt{\varepsilon_{ref}}}\left[\frac{21}{L_{s}}+\frac{65}{W_{g}}+\frac{18}{L_{g}}-3\right]$$

In step a (Figure 1), a basic circular patch antenna has been designed, resonating at 5 GHz with a return loss of −10 dB. A radius of 7 mm has been subtracted from the circle at step a; the antenna resonates at 4.8 GHz with a return loss of −10 dB. In step c, a circle is united with step b, and to obtain step d, a circle of radius 6mm is subtracted. A circle of radius 5 mm is united with the circle at step d to attain step e. Step f is obtained by attaching 4 mm radius circle to step e. Step g is obtained by uniting a circle having a radius of 3 mm. Step h is obtained by uniting a circle to step g. Step i is attained by attaching a circle with a 1mm radius to step h’s antenna design. Lastly, at step j, a 0.5 mm circle is subtracted from step i and this iteration is considered as the final one, in which we have obtained the desired results at our desired resonating frequencies.

## 3. Results and Discussions

The commercially available tool is utilized to perform the simulations with the parameters summarized in Table 1. For this entire antenna, ten iterative steps are suggested. The first step of evolution depicts a normal circular patch antenna clarified with Figure 1a. An ideal configuration of the presented crescent-shaped antenna design comprises two larger circular rings and four smaller circular ring holes embedded together, as depicted in Figure 2. The conducting patch is loaded externally with the common feed line. Figure 3 shows an iterative step of the proposed geometry. The frequency with high reflection loss is achieved. It displays unachievable impedance. This iteration process is continued for ten steps. However, in the tenth iteration, the desired operation is completed. In the ninth step in Figure 1j, two circular rings are embedded in the other larger ring. The working frequency achieved for this step is 2.8 GHz with an RL of −24 dB. In the final iteration, the frequency is achieved at the desired band, i.e., 3.1 GHz, 9.3 GHz, with a return loss of −33 dB and −32 dB. The parametric study is also observed for the inner circular ring, and the distance varies between the inner circular ring and the outer ring. Parameter d is studied for different case studies. In the final iterative step, when d is 2.5 mm, the design accomplishes a good return loss. The case studies of parameter d are displayed in Figure 4.

The obtained results of the device are tabulated in Table 2. Antenna parameter geometry against frequency with d variations is clarified in Figure 5. Here, another case study is carried out for iteration j and iteration k. In this case, minimal deviation is observed for these scenarios of an iterative process. Compared to the two bandwidth values, the second band achieves good bandwidth. The interesting feature of this work is the wider bandwidth. When the antenna operates at 3.1 GHz, it achieves 500 MHz bandwidth. Similarly, when the antenna operates at a 9.3 GHz frequency, the antenna obtains a 4800 MHz wider bandwidth. The crescent slots are realized to obtain wider impedance and good gain. The circular rings achieve good impedance, and the obtained bandwidth is utilized for many applications.

### Fabricated Prototype

The fabricated antenna of the crescent patch antenna is displayed in Figure 6. Here, in this work, half ground is considered to enhance the characteristics of the antenna. However, a minimum deviation is observed between the measurements with the simulated values. The experiments with the simulated results are evaluated in Figure 7. It is noticeable that the variation is minimal in this case. The proposed device achieves minimum return loss. The simulated model, the fabricated one in the front view, and the fabricated model in the back view with the half ground are all depicted in the figure. The crescent antenna achieves miniaturization and can be used in different applications. The crescent antenna photographs are clarified in Figure 6.

In Figure 8, the proposed design states that the E-field and H-field distributions are semi-omnidirectional and omnidirectional. The input impedance matching is also observed for the device. The proposed device achieves a 50-ohm impedance. The impedance for the presented annular ring crescent antenna is shown in Figure 9. The antenna current’s fields at 3.1 GHz and 9.3 GHz are given in Figure 10. The surface currents and electric fields are also shown in Figure 10. As shown in Figure 11, the suggested antenna has gain values of 5.8 and 4.5 over the operating frequency, which are within an acceptable range. The basic fabricated antenna with an anechoic chamber is clarified in Figure 12. As shown in Figure 13, the proposed antenna declares a radiation efficiency value of >80% across the operating frequency.

The device’s simulation current fields are also provided at two operating bands. The minimum discrepancies are observed in the experimental results. Due to manufacturing errors, there is minimal deviation. However, the simulation does not allow mismatch losses. The antenna’s bandwidth is given in Table 3. The comparison of the proposed work with the existing literature survey is listed in Table 4.

## 4. Conclusions and Future Research

Dual-band resonances are contained in an annular circular ring that is placed in the feed line, measuring 35 × 33 × 1.6 mm^3^. Optimized size and slot positioning are critical for achieving optimum impedance. The new patch conducts and produces dual-band operating frequencies. Dual-band characteristics are produced by sequentially loading the inner rings with the annular rings. The circular patch antenna was designed and simulated using ANSYS software. Due to the experimental values of dual features, compact dimension and transmission capacity are suitable for WLAN (2.8–3.2 GHz) and X-band (6.16–10.45 GHz) applications.

In the future, research on the crescent moon-shaped microstrip patch antenna with annular circular rings should focus on optimizing the design for improved performance, including increasing bandwidth and gain by incorporating advanced techniques like metamaterials and electromagnetic bandgaps (EBGs). It could expand its application to emerging technologies like 5G and IoT through further miniaturization and integration with RF circuits. Further testing, advanced simulations, and field trials will ensure durability and reliability across a variety of scenarios, while studying electromagnetic compatibility will minimize interference in densely populated frequencies.

## Figures and Tables

**Figure 1 micromachines-15-00809-f001:**
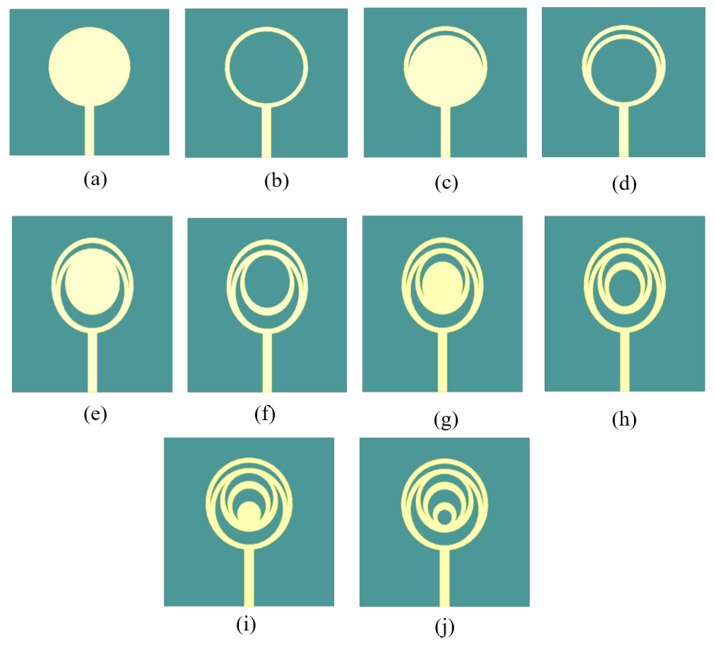
Steps to construct the crescent-moon-shaped patch antenna (**a**) iteration 1 of proposed antenna (**b**) iteration 2 of proposed antenna (**c**) iteration 3 of proposed antenna (**d**) iteration 4 of proposed antenna (**e**) iteration 5 of proposed antenna (**f**) iteration 6 of proposed antenna (**g**) iteration 7 of proposed antenna (**h**) iteration 8 of proposed antenna (**i**) iteration 9 of proposed antenna (**j**) Final proposed antenna.

**Figure 2 micromachines-15-00809-f002:**
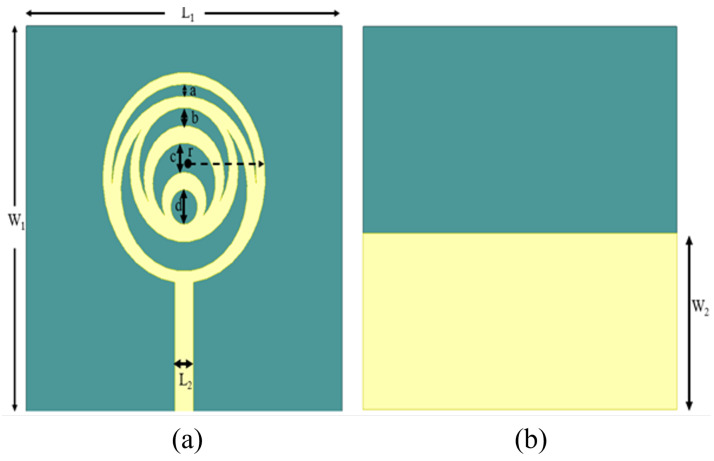
Presented antenna geometry:(**a**) top view (**b**) back view.

**Figure 3 micromachines-15-00809-f003:**
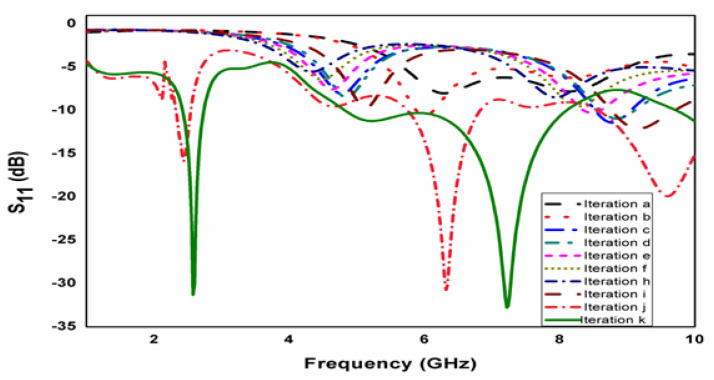
Iterative steps of proposed geometry.

**Figure 4 micromachines-15-00809-f004:**
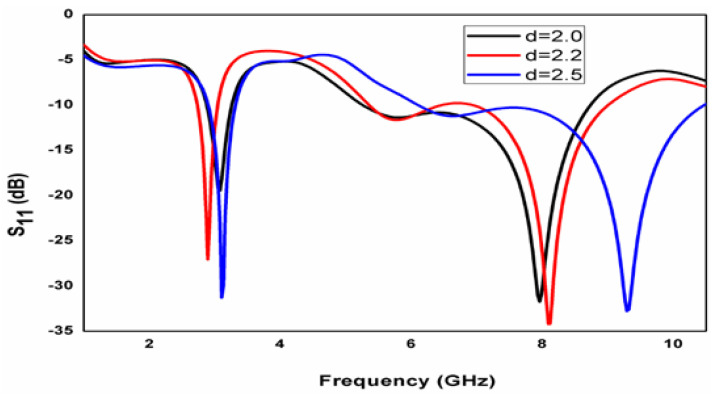
Antenna parameter geometry against frequency with d variations.

**Figure 5 micromachines-15-00809-f005:**
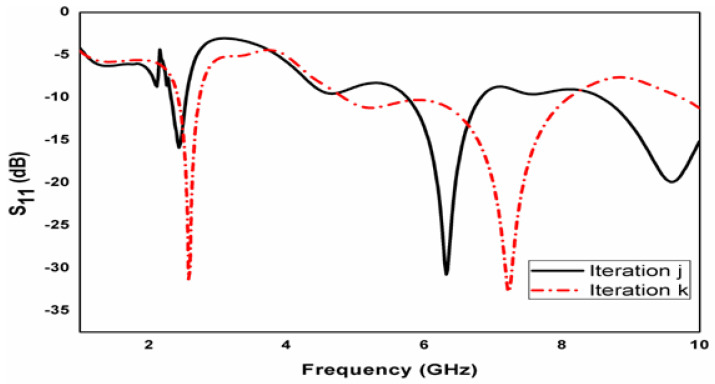
Antenna parameter geometry against frequency with iterations.

**Figure 6 micromachines-15-00809-f006:**
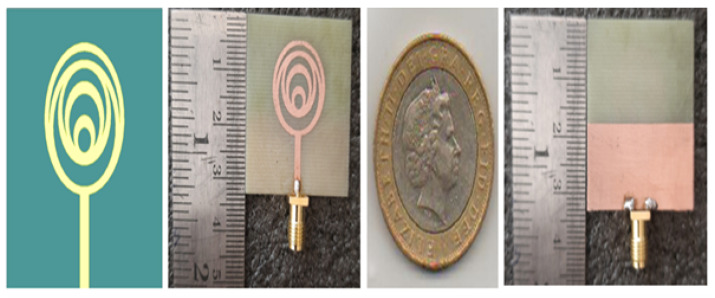
Proposed antenna fabricated model.

**Figure 7 micromachines-15-00809-f007:**
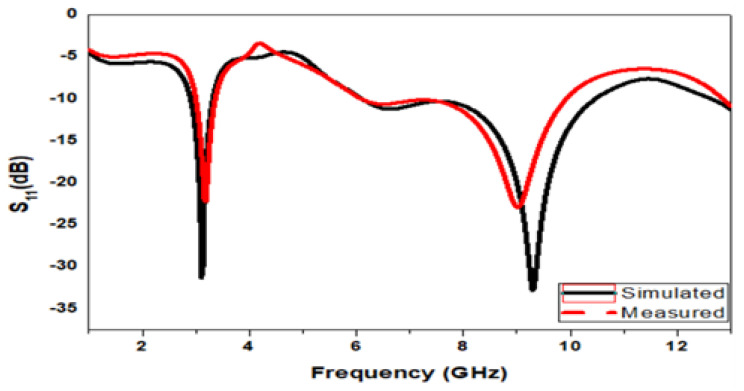
Measured with simulated results of the presented antenna.

**Figure 8 micromachines-15-00809-f008:**
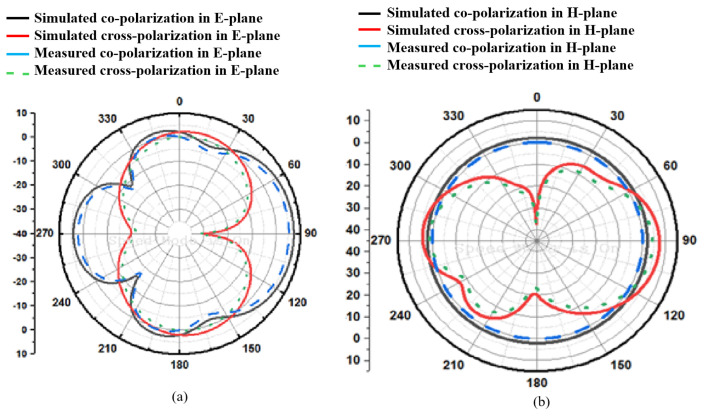
Patterns of the presented annular ring crescent antenna: (**a**) 3.1 GHz, (**b**) 9.3 GHz.

**Figure 9 micromachines-15-00809-f009:**
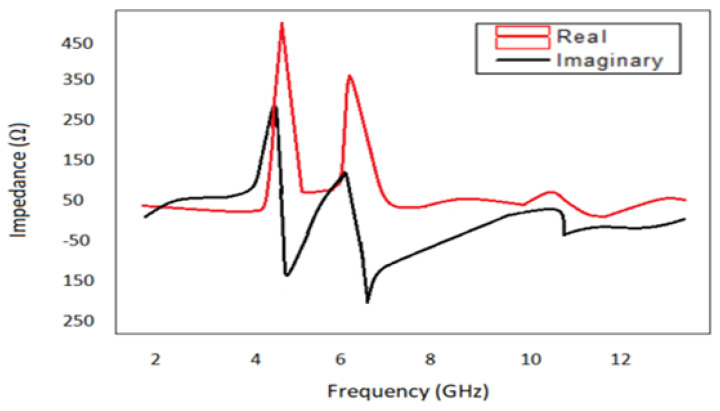
Impedance for proposed annular ring crescent antenna.

**Figure 10 micromachines-15-00809-f010:**
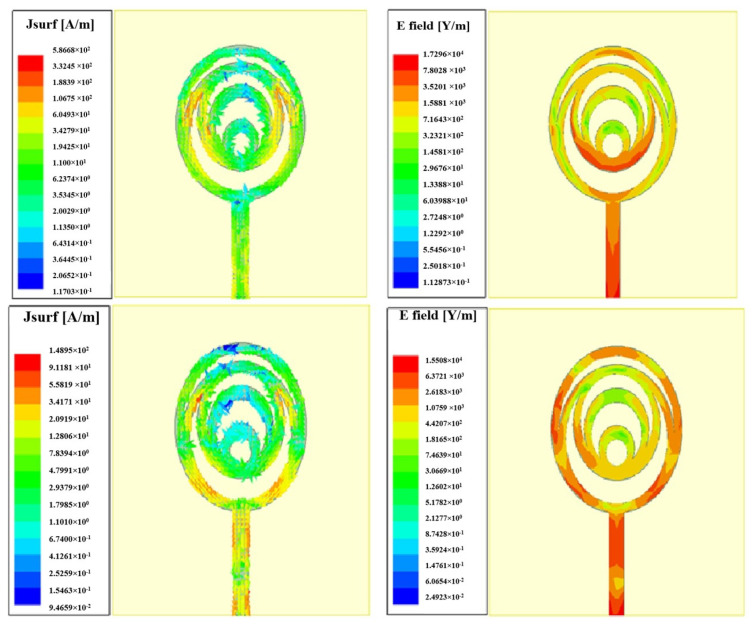
Surface current distribution of the proposed antenna.

**Figure 11 micromachines-15-00809-f011:**
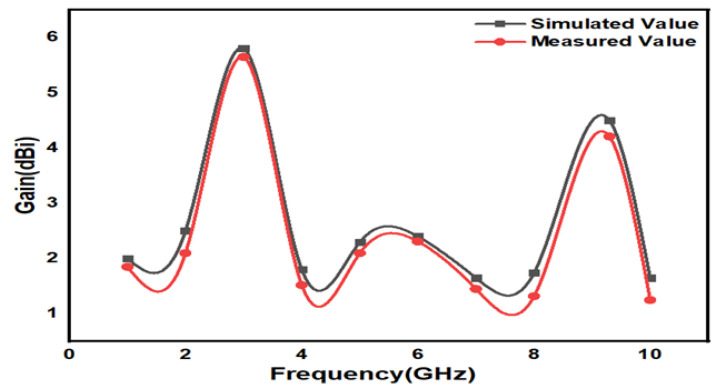
Proposed antenna gain plot.

**Figure 12 micromachines-15-00809-f012:**
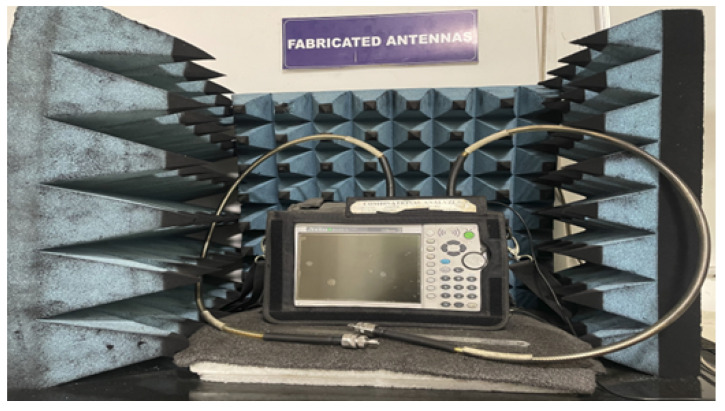
Anechoic chamber with VNA.

**Figure 13 micromachines-15-00809-f013:**
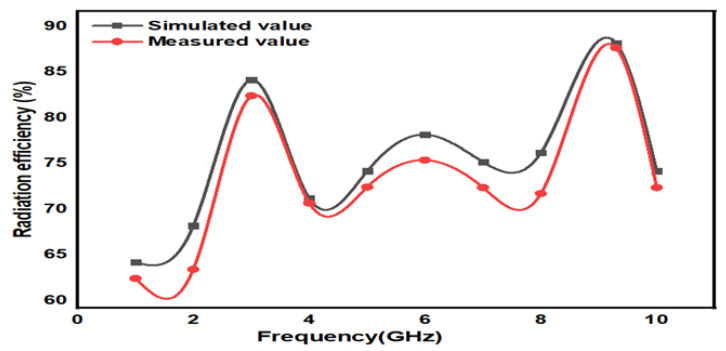
Radiation efficiency.

**Table 1 micromachines-15-00809-t001:** Presented crescent antenna dimensions.

Parameters	Dimensions (mm)
L_1_	35
L_2_	2
W_1_	33
W_2_	15
r	8
a	1
b	1.5
c	2
d	2.5

**Table 2 micromachines-15-00809-t002:** Designed antenna with measured/simulated values.

Antenna	Frequency (GHz)	Gain (dBi)
Simulated	3.1/9.3	5.8/4.5
Measured	3.15/9.38	6.1/5.4

**Table 3 micromachines-15-00809-t003:** Features of the presented device.

Antenna	Frequency (GHz)	Return Loss (dB)	Bandwidth (MHz)
Simulated	3.1/9.3	−33/−32	500/4800
Measured	3.15/9.38	−22/−24	400/4400

**Table 4 micromachines-15-00809-t004:** A comparison between the suggested work and the current literature review.

References	Dimensions (mm^2^)	Operating Bands (GHz)	Peak Gain (dBi)	Substrate Material	Design Complexity
[54]	100 × 60 × 1.6	2.17–2.72 3.34–3.66 4.85–5.77	3.75 3.56 3.93	FR4 4.2	Moderate
[55]	30 × 65 × 1.6	2.375–2.525 3.075–3.80 5.00–6.90	1.30 5.20 3.10	FR4 4.3	Moderate
[56]	40 × 45 × 1.6	2.02–2.14 4.26–4.28 5.45–5.56	1.87 2.90 4.13	FR4 4.4	Complex
[57]	33 × 50.9 × 1.6	2.80–3.00 3.30–3.50 3.80–8.00	2.32 1.21 −6.00	FR4 4.4	Moderate
[58]	40 × 40 × 1.6	2.20–2.50 4.00–4.20 6.70–8.10	NR	FR4 4.4	Complex
Proposed work	35 × 33 × 1.6	2.8–3.2 6.16–10.45	5.8–4.5	FR4 4.4	Moderate

## Data Availability

The original contributions presented in the study are included in the article, further inquiries can be directed to the corresponding authors.

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
