# Peer review of "Dual Features, Compact Dimensions and X-Band Applications for the Design and Fabrication of Annular Circular Ring-Based Crescent-Moon-Shaped Microstrip Patch Antenna"

_micromachines, 2024, doi:10.3390/mi15070809_

Round 1

Reviewer 1 Report

Comments and Suggestions for Authors

The paper titled " Dual Features, Compact Dimensions and X-Band Applications for Annular Circular Rings Based Crescent Moon Shaped Microstrip Patch Antenna Design and Fabrication" presents a patch antenna, created by inserting annular circular rings into a crescent-shaped patch.

The authors propose a detailed design algorithm on the presented antenna structure The paper introduces and describes also the iteration on the antenna dimensions.

Main contributions of the article are the detailed antenna design, the manufacturing and measurement of the proposed antenna.

Overall, the paper is well-written and organized, with clear descriptions of the design algorithm and simulation, measurement results. The work has the potential to have a significant impact on the development of WLAN antennas.

My main observations and questions are the following:

1. Please explain the bandwidth transmission, transmission capacity, transfer capacity and data transmission. These used and compared in chapter Introduction without explanations. Are these the same parameters of antennas?

2. In sentence Step 1: Using equation (1) to formulate the factor that the materials property undergoes through the Able dielectric constant (r), as follows: please correct the permittivity character.

3. Give a reference to Eq. (1) to effective permittivity, because there are numerous expressions in the literature.

4. In Eq. (5) there is an erroneous permittivity with index ref. Please correct.

5. You are using FR4 material for the antenna design and manufacturing. It worth to investigate and compare the effect of the dielectric loss on the antenna parameters at least by simulations.

Author Response

 Reviewer 1

The paper titled " Dual Features, Compact Dimensions and X-Band Applications for Annular Circular Rings Based Crescent Moon Shaped Microstrip Patch Antenna Design and Fabrication" presents a patch antenna, created by inserting annular circular rings into a crescent-shaped patch. The authors propose a detailed design algorithm on the presented antenna structure the paper introduces and describes also the iteration on the antenna dimensions. Main contributions of the article are the detailed antenna design, the manufacturing and measurement of the proposed antenna. Overall, the paper is well-written and organized, with clear descriptions of the design algorithm and simulation, measurement results. The work has the potential to have a significant impact on the development of WLAN antennas.

My main observations and questions are the following:

Comment 1-1:

1. Please explain the bandwidth transmission, transmission capacity, transfer capacity and data transmission. These are used and compared in chapter Introduction without explanations. Are these the same parameters of antennas?

Our response 1-1:

Thank you for your comment. Here in reference [27] bandwidth of 30.36% has been obtained, here we meant to say bandwidth transmission means the bandwidth of that antenna which has been designed in [27] at X-band. The term bandwidth transmission has been modified as the impedance bandwidth in the introduction part. In the introduction part transmission capacity is also referred to as bandwidth and this we have been modified in the manuscript. In [28, 29] we got an impedance bandwidth of 36.01% at X-band for both the antennas which are mentioned in the manuscript. In ref [35,36] we meant to say how much bandwidth has been obtained for both the antennas and that has been modified and highlighted in the manuscript.

Comment 1-2:

In sentence, Step 1: Using equation (1) to formulate the factor that the material’s property undergoes through the Able dielectric constant (€r), as follows:” please correct the permittivity character.

Our response 1-2:

Thank you for your comment. Here ℇeff is the effective relative permittivity of the substrate, from that we can calculate the relative permittivity of the substrate, in which we have utilized that parameter in calculating the length and width of the patch. This was modified and highlighted in the revised manuscript.

Comment 1-3:

Give reference to Eq. (1) to effective permittivity, because there are numerous expressions in the literature.

Our response 1-3:

Thank you for your comment. This is the base equation and it has been modified in the revised manuscript.

Comment 1-4:

In Eq. (5) there is an erroneous permittivity with index ref. Please correct.

Our response 1-4:

Thank you for your comment. This equation has been modified and highlighted in the revised manuscript.

Comment 1-5:

You are using FR4 material for the antenna design and manufacturing. It is worth investigating and comparing the effect of the dielectric loss on the antenna parameters at least by simulations.

Our response 1-5:

Thank you for your comment. In the comparison table we have compared FR4 substrate material with different permittivity values, by varying the permittivity of the substrate the resonant frequency may be varied. That is why we have compared our reference with [54-58] in which we have FR4 with different dielectric permittivity. Here we have chosen FR4 because it is low cost, lightweight and can be easily fabricated. As we have mentioned, if the permittivity of the substrate varies the resonant frequency may be varied. 

Reviewer 2 Report

Comments and Suggestions for Authors

The authors proposed a dual-band crescent-shaped patch antenna. The antenna evolution and results are given. Some questions are shown as follows. 

1. The explanation of performance variation of the evolution steps should be given. 

2. The average gain or gain range is better than the peak gain in Table 4. 

3. In introduction, the advantages, novelty and contribution of the proposed design should be given. 

Author Response

Reviewer 2

The authors proposed a dual-band crescent-shaped patch antenna. The antenna evolution and results are given. Some questions are shown as follows.

Comment 2-1:

The explanation of performance variation of the evolution steps should be given.

Our response 2-1:

Thank you for your comment. The explanation regarding the iteration process has been added and highlighted in the revised manuscript.

Comment 2-2:

The average gain or gain range is better than the peak gain in Table 4.

Our response 2-2:

Thank you for your comment. The gain we have mentioned in table 4 is regarding the overall gain at 2.8GHz- 3.2GHz and at X-band in which we got those from 3-db gain values.

Comment 2-3:

In introduction, the advantages, novelty, and contribution of the proposed design should be given.

Our response 2-3:

Thank you for your comment. We have added novelty and contribution to the revised manuscript according to the suggestions. 

Round 2

Reviewer 2 Report

Comments and Suggestions for Authors

The authors have well revised the manuscript. I have no more comments.